# Comparison of the Applied Measures on the Simulated Scenarios for the Sustainable Building Construction through Carbon Footprint Emissions—Case Study of Building Construction in Serbia

**Marina Nikolić Topalović [1],\*, Milenko Stanković [2], Goran Ćirović [1] and Dragan Pamučar [3]** 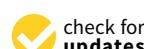

[1]  University College of Civil Engineering and Geodesy, Belgrade 1046, Serbia; cirovic@sezampro.rs
[2]  Civil Engineering and Geodesy, Faculty of Architecture, University of Banja Luka, Vojvode Stepe 77, Banja Luka 78000, Republika Srpska, Bosna i Hercegovina; milenko.stankovic@aggf.unibl.org
[3]  Department of Logistics, Military Academy, University of Defense in Belgrade, Pavla Jurisica Sturma 33, Belgrade 11000, Serbia; dpamucar@gmail.com
\*  Correspondence: marinatopnik@gmail.com; Tel.: +381-63-7772-147

**Abstract:** Research was conducted to indicate the impact of the increased flow of thermal insulation materials on the environment due to the implementation of the new regulations on energy efficiency of buildings. The regulations on energy efficiency of buildings in Serbia came into force on 30 September 2012 for all new buildings as well as for buildings in the process of rehabilitation and reconstruction. For that purpose, the carbon footprint was analyzed in three scenarios (BS, S1 and S2) for which the quantities of construction materials and processes were calculated. The life cycle analysis (LCA), which is the basis for analyzing the carbon life cycle ($LCACO_2$), was used in this study. Carbon Calculator was used for measuring carbon footprint, and URSA program to calculate the operational energy. This study was done in two phases. In Phase 1, the embodied carbon was measured to evaluate short-term effects of the implementation of the new regulations. Phase 2 included the first 10 years of building exploitation to evaluate the long-term effects of the new regulations. The analysis was done for the period of 10 years, further adjustments to the regulations regarding energy efficiency of the buildings in Serbia are expected in accordance with EU directives. The study shows that, in the short-run, Scenario BS has the lowest embodied carbon. In the long-run, after 3.66 years, Scenario S2 becomes a better option regarding the impact on the environment. The study reveals the necessity to include embodied carbon together with the whole life carbon to estimation the impact of a building on the environment.

**Keywords:** thermal insulation materials; energy rating; embodied carbon; whole life carbon; total carbon footprint; LCA

## 1. Introduction

Concern for the uncontrolled exploitation of all-natural resources together with degradation of the environment has culminated in recent decades towards increased care for the state of the environment. Consequently, all activities associated with production processes, services, transportation and construction seek possibilities to reduce the consumption of resources and impact on the environment.

On the global level, the civil engineering sector is recognized as an industry which greatly contributes to waste generation through the consumption of resources, primary materials, energy and water [1]. It is estimated that the construction sector on the global level uses 40% of energy and natural resources and produces around 33% of global emissions of $CO_2$ [2]. The need for the reduction of $CO_2$

emission by 26.90% by 2020 was identified by Kim et al. [3]. The studies, often quoted as examples of the impact on the environment from the construction sector [4], show that the use of cement is responsible for 8.6% of $CO_2$ emission in the world [5]. Cement is widely used in civil engineering, both to produce concrete and as matrix for many other products which are used on global level. One of the basic concepts of the economics of natural resources and environment is sustainability and sustainable development [6], especially sustainability of transportation and logistics [7–11]. In relation to sustainability, EU commission [12] decided that it is necessary for the civil engineering sector to implement measures to reduce emissions and mitigate climate changes. The measures refer to the savings in using primary materials, using recycled materials, reducing waste quantity, returning materials into energy cycles, and reducing water and energy consumption. This also includes the studies on possible savings in energy consumption in buildings [13–16]. According to Vourdobas [17], energy consumption in buildings accounts for 40% of total energy consumption in EU.

Recommendations about the ways to reduce energy consumption and implement the measures for energy efficiency of the buildings are given in the approved EU Directive (Directive2002/91/EC and Directive 2010/31/EU) [18,19]. Serbia's legislations are trying to follow EU Directive and reduce operational energy in buildings through energy efficiency measures, by introducing energy passports for new and existing buildings [20,21]. Accordingly, since September 2012, Serbia has laid down a minimum condition for new buildings, i.e. the energy rating C. Methodology for calculation of the energy requirements of a building, according to the rules of energy efficiency of buildings, in compliance with EU Directive [18,19], is based on the calculation of operational energy [20] (energy used for comfortable use of a building) and that consumption is the basis of the classification of buildings into energy ratings. The embodied carbon in buildings is not included in the calculation, but its value increases with the implementation of the new measures on energy efficiency, due to the increased flow of thermal insulation materials.

## 2. Overview of the Reference Papers Used for the Research

The importance of improvement of energy efficiency through the reduction of operational energy was emphasized by Vourdoubas [17]. Certain activities are undertaken to increase the thickness of thermal insulation materials in thermal layers of a building which consequently results in the rise in the embodied energy, i.e., embodied carbon in a building. That embodied carbon particularly is to be included in the calculation of the impact a building has on the environment [22]. The current European regulations give directions for the designers to design energy efficient buildings with zero net energy consumption [17], the goal of which is that by 2020 all public buildings are to be zero-energy buildings; however, the focus is on operational energy. The total carbon footprint from the construction sector is measured based on operational energy, which is the basis for forecasts and plans for the emission reductions on both local and global levels. Vourdoubas [17] suggested that zero-energy buildings, despite their zero-energy consumption in exploitation phase, do impact the environment, which consequently comes from those higher demands and that a concept of zero $CO_2$ emission, from operational and embodied energy, has not been developed so far. The same author concluded that there are no regulations for zero-energy buildings.

The impact a building has on the environment starts with the exploitation of natural materials from the environment (cradle) followed by the production of construction materials, transportation, construction and continues through the building life span, its maintenance, replacement of components and at the end of the life cycle its demolition and recycling or generated waste. The total emissions of the building life cycle are the outcome of operational and embodied energy, as asserted by Vourdoubas [17].

EU Directives (Directive 2002/91/EC and Directive 2010/31/EU) [18,19] focus on the energy used in the operational phase (operational energy), while Vourdoubas [17] suggested that, along with operational energy, which is used during the building exploitation, embodied energy used during the construction of a building as well as during the production of the building materials, is usually not taken into consideration. Ibn-Mohammed et al. [23], who also reviewed the current literature on

the embodied energy in a building, stated that, for mitigating climate changes, buildings should be designed and constructed with the minimum effect on the environment.

In a study carried out on a house in Denmark [24], it is assumed that the construction phase accounts for 5–11% of the total energy consumption until the exploitation phase. Analyzing input and output [25], it is estimated that 6% of energy is consumed during the construction of a building compared with the energy of the building life cycle, while some authors [26] claim that its share is negligible compared with the energy consumed in exploitation during the building life span. Nässén et al. [25] concluded that the differences in energy consumption estimation occur because of the different system boundaries. Certain authors consider the exploitation phase of a building and equipment [27] responsible for consumption of most energy and consequently has the highest production of $CO_2$. Ramers et al. [27] analyzed the results of 73 cases of office and residential buildings in 13 countries. They ascertained that the operational energy, during the building life cycle, was 80–90% and the rest was embodied energy. The impact of Phases A1–A5 on the environment in the life cycle of a building wasstudied by the authors who concluded that this phase has aproportion of 8–20% [28–36]. The impact of thermal covers of a building on the environment in Phases A1–A5 was investigated by Gámez-García et al. [37].

The applied estimation of annual consumption of primary energy was 150–400 kWh/m$^2$ for residential buildings and 250–550 kWh/m$^2$ for office buildings in Serbia [38]. Analyzing the operational end embodied energy of a building in Italy, Cellura et al. [39] statedthat a key question is the embodied energy of a building and concluded that it is especially important for low-energy buildings.

In recent studies on energy efficiency of buildings, Karimpour et al. [40] concluded that, in mild climates, the embodied energy in buildings can account for about 25% of total life cycle energy. The same authors believed the trend for designing and constructing zero-energy buildings will lead to the rise in embodied energy as well as the total building life cycle energy. Their studies [40] show that the current estimations of embodied energy are unclear and differ considerably, thus they conclude that guidelines are necessary, which would develop into globally accepted protocol for the calculation of embodied energy. The extensive study carried out by Gámez-García et al. [37] on the impact of the applied materials in external walls of residential buildings in Spain shows the advantages and disadvantages of certain assemblies and negative impacts of gypsum boards. They claimed that 90% of the impact is generated during the process of production, whereas the process of construction solely contributes 1% of the impact [37].

Karimpour et al. [40] claimed that it is also necessary to consider embodied energy and not only operational energy, since the impacts on the environment are directly associated with energy consumption. Fay et al. [41] concluded that the embodied energy has become important and it is necessary to consider its impacts as well. Recent studies done by Battle et al. [42] and Sturgis [43] also indicate the need for analyzing the embodied carbon and comparing it with the whole life carbon in a building, and they concluded that it is also necessary to calculate the carbon footprint in installed materials.

The studies on the impact of the products used in the construction of a building by applying life cycle analysis (LCA) can help when deciding what product or system [44,45], planned for the building construction, to choose. Battle et al. [42] indicated how important the analysis of carbon footprint is throughout the whole life cycle by studying the relationship between the emissions of embodied and whole life carbon footprint in a Deloitte building whose life span is 100 years. The study results show that the level of embodied carbon is much higher than it has been previously assumed.

In Serbia, studies on the impact of civil engineering sector on the environment are based on the studies of the exploitation phase of a building, such as the research done by Krstić-Furundžić et al. [46] and Grahovac et al. [47] as well as the others [38,48]. The aim of the European project Tabula, which involved an expert team from Serbia, was to analyze the existing building stockand suggest measures for its energy rehabilitation that would reduce consumption of operational energy [49].

The importance of the impact of the applied construction materials through the analysis of the life cycle of the materials was also addressed by local authors. Slavković et al. [50] and Jovanović Popović et al. [51] stated that, for further energy saving, investigating the embodied energy in materials during their life cycle is necessary. Jovanović Popović et al. [49] noticed that, despite the ample work done in Tabula project, the ecological aspect of the applied materials was left unexplored. They had found this by applying life cycle analyses of the materials, which include estimation of the embodied energy "from cradle to cradle", through which it is possible to improve the construction fund, which would consequently lead to energy savings.

In that regard, the aim of this research was to calculate the embodied carbon in Phase 1 and the total carbon footprint in Phase 2 on the residential building projects. These two phases of the research show the short- and long-term effects of the applied measures for energy efficiency of buildings, respectively, according to the legislation on energy efficiency which has been in use in Serbia since 2012 and follows EU Directives (Directive 2002/91/EC and Directive 2010/31/EU) [18,19].

The aim of the first part of the research was to estimate the value of the embodied carbon for the analyzed scenarios. In that regard, the first part of the research was conducted with the aim to identify how more embodied carbon is created by thermal insulation materials and façade carpentry with the improved technical features, as well as to quantify that difference.

In the second part of the research, the total carbon footprint was calculated after 10 years of operation of the building, and the obtained results were compared between the analyzed scenarios. The aim was to estimate the total impacts of the construction and exploitation of the building and assess when the positive effects of the applied measures on energy efficiency of buildings can be expected.

The model of building management shown in this paper is different from the usual way of calculating energy rating of a building according to the legislation in Serbia [20,21] because the calculation also includes embodied carbon and not only whole life carbon, as determined by the current methodology. This calculation of the impact of a building on the environment, which also includes the construction phase and not only the operational phase of the building, gives a clear image of the real impacts of a building on the environment. This provides a realistic estimation about when the positive effects can be expected, because of the implementation of the measures for the energy efficiency of buildings in Serbia. The suggested model of calculation can be applied in other EU countries by analyzing the embodied carbon on typical projects of those countries, as was investigated by Gámez-García et al. [37] for the external walls used in Spain.

## 3. Methodology

The design for the environment, LCA instruments and manufacturers' responsibilities are the key tools used by prosperous companies to develop and improve the characteristics of the existing products and services [52]. The purpose of "green design of products and services" is to recognize, identify, estimate and minimize the impacts of the products and services on the environment. This is achieved by systematic examination of the products' characteristics through the analysis of the impact on the environment, health and safety of the population during the products' life cycle, i.e., starting from the design phase, production, exploitation to final disposal. Life cycle comprises consistent and interconnected phases of a product production, from raw materials or natural resources (cradle) to final disposal (grave) [53]. The European Commission has recognized LCA as a methodology for identifying environmental intervention and potential impacts which a product or service has during its life cycle [54]. Accordingly, the International Organization for Standardization (ISO) specified LCA methodology in ISO 14040:2006 and 1044:2006 [55,56].

This research, aiming to assess the value of the embodied carbon generated by the increase in the flow of thermal insulation materials and its participation in total carbon footprint, was conducted in two phases. The system boundaries are in accordance with the research goal. Step 1 calculated the carbon footprint in the baseline scenario (the building without thermal insulation) and compared with the carbon footprint of Scenarios S1 and S2. Step 2 was the inventory analyses. The inventory comprises

materials, activities and energy sources which are calculated from the design plan, i.e., bill of quantities by using The Norms and Work Standards in Civil Engineering [57]. The precisely calculated quantities of materials, activities and energy sources for each analyzed scenario are the life cycle inventory within the system boundaries. The quantities obtained in that way were put into Carbon Calculator, in which it is possible to choose the type of transportation and the distance from where the materials are supplied. Step 3 was the impact assessment, which was expressed through construction carbon footprint (embodied carbon from the first part of the research or the total carbon footprint from the second part of the research). To reach the improvement, the impressions were analyzed, and conclusions and recommendations were made. In Step 4, conclusions were drawn regarding the advantages of each scenario as well as when decreased impact on the environment can be expected. In the short-run, the embodied carbon calculated in Phase 1was within the boundaries of Phases A1–A5, whereas Phase 2, within the boundaries of Phases A1–B2 (after 10 years of exploitation), gave long-term results.

LCA was used as the basis for $CO_2$ emissions calculation. The analysis of the embodied carbon footprint of a building is a methodology which relies on the principles of measuring life cycle performance of a building and the purpose of which is to calculate embodied carbon and whole life carbon. Some authors [42,43] concluded that calculation of the carbon footprint in the design stage can contribute to the decrease in the impact of civil engineering sector on the environment [37].

Elaborate research on embodied carbon in the external walls of the residential buildings, where LCA methodology was applied, shows that it is possible to choose the structure of the walls together with accompanying components as well as the advantages and disadvantages of certain layers [37]. The importance and implementation of LCA are discussed by various authors [58–63].

The boundaries of Phase 1 to calculate embodied carbon according to the standard EN 15978: 2011 [64] are of Phases A1–A5. The system boundaries are set in compliance with the goal of this part of the research. The aim was to investigate whether there are differences in the values of embodied carbon in these three scenarios, and conclude which scenario is the most beneficial for the environment. Quantifying the embodied carbon in these three scenarios would clarify the level of additional impacts on the environment from the moment the measures for energy efficiency of buildings are implemented according to the legislation in Serbia which came into force in September 2012 [20,21].

In Phase 2, the boundaries of the conducted research are according to standard EN 15978: 2011 [64] of Phases A1–B2 (after the first 10 years of exploitation). According to some authors [42], maintenance and replacements of certain components start after 10 years of exploitation, which consequently causes new emissions of embodied carbon, which are not the subject matter of this research. One reason the boundaries are set in this way was due to the lack of legislations concerning material replacement. The other reason was the expected changes in the legislations in Serbia and their adjustments with EU directives.

The boundaries of the system to estimate total carbon footprint including 10 years of exploitation of the building were applied as a methodological approach in LCA to follow the flow of resources, energy, emissions and waste. This approach enables the comparison between the scenarios and decision making. Databases are the sources of information to analyze inputs and outputs in LCA. Since Serbia does not have available public data on the national base of materials and products, the database used for this research is ICE version 2: Inventory of Carbon and Energy [65]. ICE database studies the concept of implemented materials with reference to their embodied energy and embodied carbon, in broader scope of GHG (Greenhouse Gases), building components and transportation. The Environment Agency UK Carbon Calculator [66] was used for analyzing carbon footprint. This software has a database of materials produced from natural raw materials (primary materials), because the products made of primary materials are used in the construction industry in Serbia [67]. The availability of the software Carbon Calculator is an additional reason for its use in this research. The program URSA construction physics 2 [68] was used to calculate the building energy rating based on which the operational carbon was measured only for the heating of the building.

This study shows how it is possible, by calculating the embodied carbon in the design stage, to estimate the level of impact on the environment which comes from the implementation of the measures for energy efficiency of buildings to achieve energy rating C or B. Apart from that, this study shows how it is possible, by calculating the embodied carbon in the design stage, to affect short- and long-term policies for the reduction of carbon footprint in civil engineering sector on the national level and investor level.

The research was conducted on the project of a residential building with a net area of 110 m² close to Belgrade. For this purpose, three scenarios were made: Scenario S1, the house is designed for energy rating C; Scenario S2, the house is designed for energy rating B; and Scenario BS (baseline), the house does not have any thermal insulation layers. Input data, such as the quantities of construction materials, energy sources for the machines, duration of construction, inventory for LCA analysis, were calculated from the bill of quantities by means of the Norm and Work standards in Civil Engineering [57]. Energy rating and the sizing of thermal cover for all three scenarios were done in program URSA construction physics 2 [68], to enable the correct calculation of the required quantities of materials as well as the necessary energy for heating on annual level. The planned energy source in all three scenarios in operational phase was pellet, which is locally produced and available.

## 4. Case Study

In 2008, before the legislation in Serbia was approved, energy efficiency in building construction in Serbia was affected by the participation of building sector with 41% of energy consumption [69]. The production and consumption of energy is directly connected with the production of $CO_2$ and, according to the available data, there was a drop in energy consumption and $CO_2$ production in 2013, but still it is the highest in the region amounting to 6.33 tons $CO_2$/capita [70]. In 2013, the national ecological footprint in Serbia was 3.02 global hectares [71]. More than 50% of the ecological footprint in Serbia comes from the production of $CO_2$ [71]. Serbia has been making efforts to reduce energy requirements in building constructions by implementing certain measures since 2012 on both new and existing buildings through energy rating and rehabilitation. Thus far, 1600 energy passports have been issued in Serbia for both new buildings and energy rehabilitations. Approximately 98.5% of those passports are for energy rehabilitation of existing buildings with lower ratings and for new buildings with energy rating C, and only about 1.5% are the buildings with higher energy ratings B and A.

The valid methodology for calculating the energy efficiency of buildings in Serbia came into force in 2012, trying to reduce the impact of buildings on the environment. That methodology for calculating energy and energy rating considers only the energy consumed in the exploitation phase, but the impacts generated in Phases A1–A5 are neglected according to the valid methodology for calculating the impact of a building on the environment in Serbia. The impact of the increased flow of thermal insulation materials and the carpentry with the improved features will be included in the energy rating of buildings through the new regulations on energy efficiency of buildings. In this research, the impact in Phases A1–A5 was quantified and, together with the exploitation phase of a building, they can measure when the reduced impact of a building on the environment can be expected, based on which the energy rating of a building is decided. The research shows that it is necessary to estimate the total carbon footprint of the construction even in the design phase and decide the energy rating of a building based on that finding. In the long-run, this type of project management enables the management of ecological footprint as well as the planning of the activities to mitigate the national ecological footprint in Serbia.

*Description of the Experiment*

The experiment was done on a building plan for a four-member family ground floor house with a net area of 110 m² near Belgrade. The house was designed, according to the detailed regulation plan, on the location which is intended for the residential buildings with access roads and connections to electricity and water supplies and sewerage system. According to the National typology of residential

buildings in Serbia, Jovanović Popović et al. [49] found that the family residential buildings are represented with 96.3% compared to multifamily residential buildings with 3.7%. This is the reason for choosing the family residential building for this research.

The experiment which measures the embodied carbon was performed on the building plan for each analyzed model for the family house built with primary materials. Scenario BS is without thermal insulation layers in thermal covers and with poor quality façade carpentry, which was the usual way of construction in Serbia before the new regulations. However, Scenarios S1 and S2 have thermal insulation layers on the thermal cover and facade carpentry with the improved thermal characteristics. The foundation plan of the building is shown in Figure 1. The side elevation of the building from the direction of the street is shown in Figure 2.

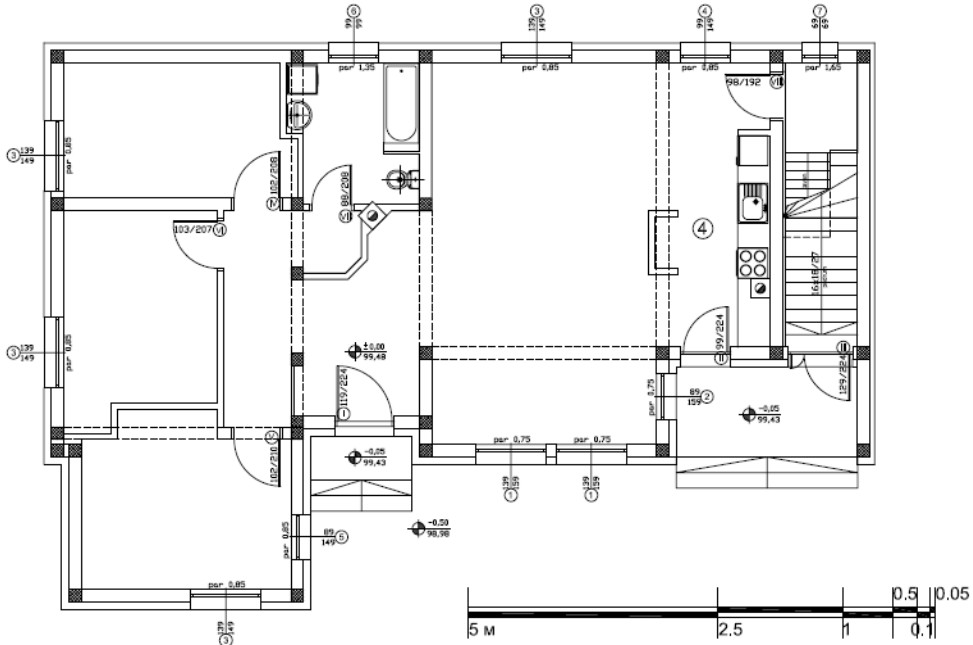

**Figure 1.** Foundation plan of the building.

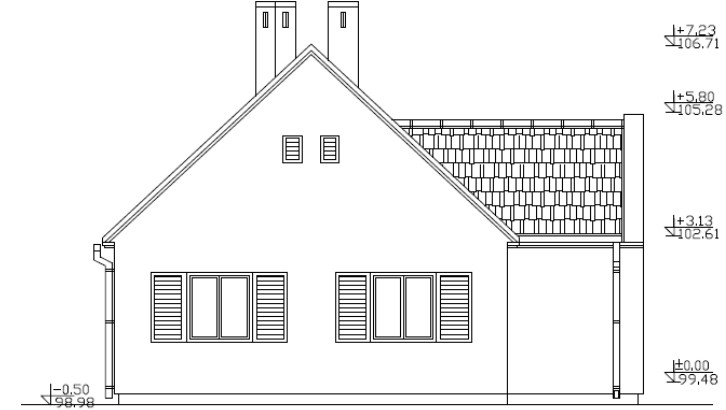

**Figure 2.** Side elevation of the building from the direction of the street.

The quantity of materials and the structure of thermal cover are based on the Norm and Work standards in Civil Engineering [57] including all transportation routes from the factory, duration of the construction and the waste generated out of these activities, as well as the energy sources and water for all three scenarios. Program URSA construction physics 2 was used for the thermal calculation of Scenarios S1 and S2. Scenario S1 is the house designed for energy rating C, and Scenario S2 is the house designed for energy rating B according to the legislation in Serbia [20,21]. The products and

materials used for the construction were from the local construction industry. All transportation routes were calculated for each position as well as the modes of transportation: road haulage, tow trucks and lorries from the factory to the building site, and vans for smaller quantities. The work force for the construction was engaged from within 30 km. The duration of construction was 16 weeks, which is in compliance with the Norm and Work standards in Civil Engineering [40], and the calculation of the carbon footprint included, in addition to the required construction materials and products, activities of construction, work force accommodation, generation of hard municipal waste, water, electricity, and fuel consumption for devices and machine son the building site to measure the impact of the analyzed Scenarios S1 and S2 as well as Scenario BS.

The program URSA construction physics 2 [68] was used to calculate the building energy rating and measure the thermal cover, whereas ICE database [65] (Inventory of Carbon and Energy, ICE) and software from Environment Agency UK [66] were used to calculate embodied carbon. The analyzed project is a family house with a net area of 110 m$^2$ and the materials used for its construction are in quantities obtained by designing and applying the Norm and Work standards in Civil Engineering [57]. Data on required materials, processes, energy sources and products are shown in Table 1.

**Table 1.** Quantity of materials and energy sources used for each scenario.

| Type of Material and Energy Source | Units of Measure | Replaced Quantities | | |
|---|---|---|---|---|
| | | **BS** | **S1** | **S2** |
| Tamping gravel | (m$^3$) | 75.00 | 75.00 | 75.00 |
| Crown tile | (pc) | 10.240 | 10.240 | 10.240 |
| Bricks and clay blocks, easy installed ceiling | (m$^3$) | 92.00 | 92.00 | 92.00 |
| Cement mortar | (m$^3$) | 23.40 | 23.40 | 23.40 |
| Lime mortar | (m$^3$) | 7.80 | 7.80 | 7.80 |
| Steel reinforcement | (tons) | 6.50 | 6.50 | 6.50 |
| Concrete MB30 | (m$^3$) | 38.00 | 38.00 | 38.00 |
| Concrete MB20 | (m$^3$) | 62.50 | 62.50 | 62.50 |
| Ceramic tiles | (m$^2$) | 87.00 | 87.00 | 87.00 |
| Glue for tiles and parquet | (kg) | 490 | 490 | 490 |
| Lacquer for parquet | (L) | 30 | 30 | 30 |
| Total amount of timber | (m$^3$) | 18.70 | 18.70 | 18.70 |
| Parquet or match floor | (m$^3$) | 3.10 | 3.10 | 3.10 |
| Thermal insulation, polystyrene | (m$^3$) | - | 37.50 | 67.50 |
| Thermal insulation, mineral wool | (m$^3$) | - | 21.50 | 35.83 |
| Thermal insulation, austrotherm | (m$^3$) | - | 14.00 | 21.00 |
| Facade mortar | (kg) | 800 | 800 | 800 |
| Interior paint for walls | (kg) | 100 | 100 | 100 |
| Mass for skimming | (kg) | 500 | 500 | 500 |
| Window glass | (m$^3$) | 0.60 | 0.80 | 1.21 |
| Electrical installation | (kg) | 520 | 520 | 520 |
| Heating installation | (kg) | 750 | 750 | 750 |
| Waterworks and sewage works | (kg) | 150 | 150 | 150 |
| Roofing paper | (kg) | 150 | 150 | 150 |
| Hydro insulation | (m$^3$) | 1.50 | 1.50 | 1.50 |
| Personal transportation within 30 km | (km) | 5.400 | 5.760 | 5.820 |
| Transportation of waste to landfill | (m$^3$) | 110.00 | 112.00 | 112.00 |
| Water consumed on site | (L) | 20,600 | 20,800 | 20,800 |
| Power consumed on site | (kWh) | 13,500 | 13,500 | 13,500 |
| Diesel fuel consumed on site | (L) | 900 | 900 | 900 |

The system boundaries in Phase 1 are of Phases A1–A5 to calculate the embodied carbon which is generated from that phase of the building, without operational phase. The impacts from the exploitation phase are beyond the system boundaries and were not included in the Phase 1 calculation. The impact of each scenario is presented by the carbon footprint $CO_2$e. ($CO_2$ equivalent as

a total measurement of carbon impact and all other greenhouse gases, processes, and transportation). To compare carbon footprint of Phases A1–A5, Scenarios S1 and S2 were made, which were compared with Scenario BS. The comparison of Scenarios S1 and S2 with the referential Scenario BS, which is without thermal insulation materials, was made to observe the relative impact on the environment upon implementation of the regulations on energy efficiency of buildings in Serbia [20,21].

The system boundaries in Phase 2 are of Phases A1–A5 and include the first 10 years of operational Phase B2 to calculate the total carbon footprint which is generated from cradle A1 during production, transportation and building construction and B2 in the first 10 years of the exploitation phase, without maintenance, replacements, mending and reconstruction from operational, end-of-life and reuse/recycling phase. The impacts from maintenance, repairing and replacement are the parts of the operational phase which, after 10 years of exploitation, are the cause of the increase in embodied carbon, thus are beyond the system boundaries and were not included in the calculation, because changes in the regulations on energy efficiency of buildings, adjusted to the EU regulations, are expected. The boundaries set in this way help estimate the long-term, i.e., 10-year, consequences and impacts on the environment from the moment of applying measures "for making buildings warmer" according to the regulations on energy efficiency and building certification in Serbia [20,21]. The impact of each scenario is presented by the carbon footprint $CO_2e$.

The program [66] used to calculate carbon footprint has the possibility to adjust the modes of transportation, distances, the way of waste management, the amounts of the used energy and generated waste, and the duration of construction. The calculation of carbon footprint of materials and activities participating in construction is the sum of emissions of each material, activity and energy source, as shown in Table 1. The main variable factors are: the quantity of thermal insulation materials and materials for façade carpentry with the improved technical features, the amount of necessary transportation of equipment and removal of materials, and personal transportation. The embodied carbon is shown for the groups of materials and processes, thus it is possible to calculate the amount of $CO_2e$ for each scenario to estimate embodied carbon, and, in Phase 2, the amount of whole life carbon.

Scenario BS is a building designed in load bearing structural system. The walls are from hollow brick blocks 25 cm thick, plastered on both sides, with RC (reinforced concrete) columns, horizontal and vertical ring girders, easily installed ceiling bricks 16+4, and lightweight reinforced floor slab. Floor and wall finishing are in accordance with the purpose of the room. Windows, the front door and cloth blinds are regular and with poor technical features. The building is without additional thermal insulation layers, and the calculation shows that it is in energy rating G. The details of thermal layer in Scenario BS (external wall, external carpentry, ceiling and floor) are shown in Figure 3. The parameters of thermal layer, crucial for the calculation of energy rating of Scenario BS (external wall, external carpentry, ceiling and floor), are shown in Table 2.

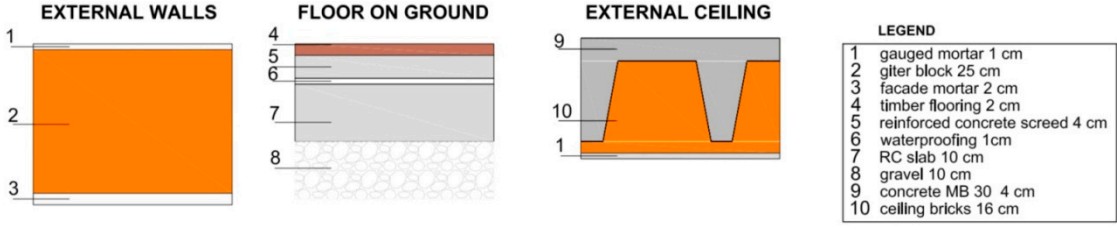

**Figure 3.** Details of thermal layers in Scenario BS.

Scenario 1 (S1) is a building designed in energy rating C and designed in load bearing structural system. The walls are from hollow brick blocks 25 cm thick with 12 cm of thermal insulation on the facade walls with decorative external plaster and internal gauged mortar, with RC columns and the horizontal and vertical ring girders, easily installed ceiling bricks 16+4 concrete, with 15 cm thick thermal insulation towards the attic. Lightweight reinforced floor slab is covered with 10 cm thick thermal insulation, cement screed and the floor finishing are in accordance with the purpose of the

room. Facade carpentry with the improved technical features is in accordance with the minimum requirements according to the new regulations necessary to achieve energy rating C. The details of thermal layers in Scenario S1 (external wall, external carpentry, ceiling and floor) are shown in Figure 4. The parameters of thermal layer, crucial for the calculation of energy rating of Scenario S1 (external wall, external carpentry, ceiling and floor), are shown in Table 2.

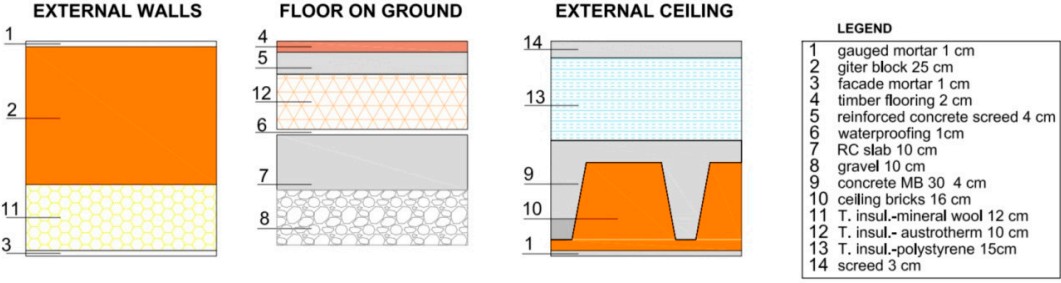

**Figure 4.** Details of thermal layers in Scenario S1.

Scenario 2 (S2) is a building designed in energy rating C and designed in load bearing structural system. The walls are from hollow brick blocks 25 cm thick with 20 cm of thermal insulation on the facade walls with decorative external plaster and internal gauged mortar, with RC columns and the horizontal and vertical ring girders, easily installed ceiling bricks 16+4 concrete, with 25 cm thick thermal insulation towards the attic. Lightweight reinforced floor slab is covered with 18 cm thick thermal insulation, cement screed and the floor finishing are in accordance with the purpose of the room. Facade carpentry is with the improved technical features necessary to achieve energy rating B. The details of thermal layers in Scenario S2 (external wall, external carpentry, ceiling and floor) are shown in Figure 5. The parameters of thermal layer, crucial for the calculation of energy rating of Scenario S2 (external wall, external carpentry, ceiling and floor), are shown in Table 2.

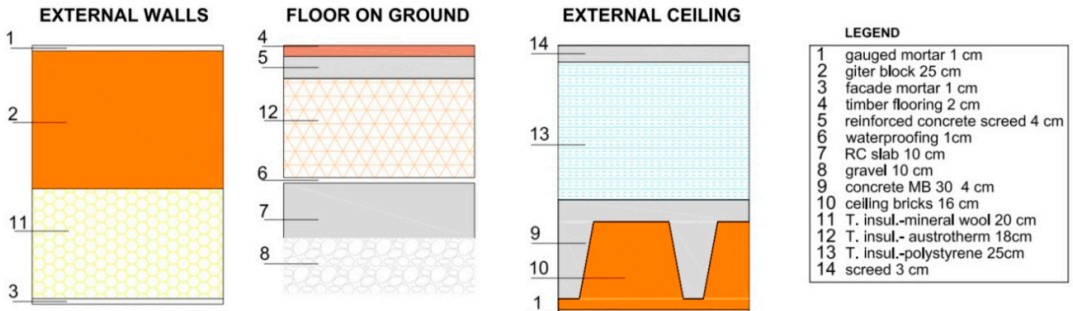

**Figure 5.** Details of thermal layers in Scenario S2.

The three analyzed scenarios have the same constructive systems which differ in the thickness of the thermal insulation materials, installed in the thermal covers, and thermal features of the external carpentry.

**Table 2.** Parameters of thermal layer, crucial for the calculation of energy rating.

|  | Scenario BS | | Scenario S1 | | Scenario S2 | |
| --- | --- | --- | --- | --- | --- | --- |
|  | U-Value (W/m$^2$ K) | Thickness (cm) | U-Value (W/m$^2$ K) | Thickness (cm) | U-Value (W/m$^2$ K) | Thickness (cm) |
| External wall | 1.46 | 28 | 0.24 | 39 | 0.153 | 47 |
| External carpentry | 2.90 | 6 | 1.49 | 7 | 0.90 | 8 |
| External ceiling | 1.92 | 21 | 0.238 | 40 | 0.15 | 49 |
| Floor | 3.00 | 17 | 0.30 | 27 | 0.18 | 35 |
| Energy rating | **G** | | **C** | | **B** | |

## 5. Results and Discussion

### 5.1. Results of the First Part of the Study for Scenarios BS, S1 and S2 in Phase A1–A5

The results of the embodied carbon footprint for the analyzed scenarios were obtained after Phase 1 was completed. The amount of embodied carbon footprint for the groups of materials and activities participating for Phases A1–A5 of Scenario BS and analyzed Scenarios S1 and S2 are shown in Table 3. To get the results of the carbon footprint of the construction $CO_2e$ (embodied carbon), the operational phase is not included in the calculation as it is outside the system boundaries in Phase 1.

**Table 3.** Values of carbon footprint in the construction phase (embodied carbon) and percentage of participation in analyzed scenarios (BS, S1 and S2).

| Groups of Materials and Activities | BS | | S1 | | S2 | |
|---|---|---|---|---|---|---|
| | Tons $CO_2e$ | Percent | Tons $CO_2e$ | Percent | Tons $CO_2e$ | Percent |
| Quarried material | 44.40 | 32.08 | 44.40 | 29.96 | 44.40 | 29.01 |
| Timber | 3.40 | 2.46 | 3.40 | 2.29 | 3.40 | 2.21 |
| Concrete, mortars &cement | 28.40 | 20.52 | 28.40 | 19.16 | 28.40 | 18.55 |
| Metals | 23.90 | 17.27 | 23.90 | 16.13 | 23.90 | 15.61 |
| Plastics | 5.80 | 4.19 | 5.80 | 3.91 | 5.80 | 3.79 |
| Glass | 1.40 | 1.01 | 3.70 | 2.50 | 5.60 | 3.65 |
| Miscellaneous | 1.60 | 1.16 | 9.00 | 6.07 | 11.90 | 7.77 |
| Finishing, coatings & adhesives | 7.10 | 5.13 | 7.10 | 4.79 | 7.10 | 4.64 |
| Plant and equipment emissions | 5.40 | 3.90 | 5.40 | 3.65 | 5.40 | 3.52 |
| Waste removal | 1.10 | 0.80 | 1.10 | 0.74 | 1.10 | 0.72 |
| Portable site accommodation | 2.00 | 1.45 | 2.00 | 1.35 | 2.00 | 1.31 |
| Material transport | 5.50 | 3.96 | 5.60 | 3.78 | 5.70 | 3.73 |
| Personnel travel | 8.40 | 6.07 | 8.40 | 5.67 | 8.40 | 5.49 |
| **Operational** | 0.00 | 0.00 | 0.00 | 0.00 | 0.00 | 0.00 |
| **Total Carbon Footprint** | 138.40 | 100.00 | 148.20 | 100.00 | 153.00 | 100.00 |

### 5.2. Discussion and Comparison of Scenarios BS, S1 and S2 upon the Completion of the Analysis of Embodied Carbon for PhasesA1–A5

The values of embodied carbon benchmarks for Phase 1 are given in Table 4. The value of the embodied carbon in Scenario BS is 138.40 tons $CO_2e$ for the whole building, which is the lowest value of all three scenarios. When comparing values per square meters ($m^2$) of net area of the building in Scenario BS, the obtained result is 1.26 tons $CO_2e/m^2$. Based on the amount of embodied carbon, Scenario BS is the most favorable regarding the impact on the environment. The next least suitable scenario for the environment, in Phase 1, is Scenario S1 whose embodied carbon is 148.20 tons $CO_2e$, for the whole building, which is 9.80 tons $CO_2e$ more than that in Scenario BS with an increase of 7.08%. The obtained value calculated per $m^2$ for Scenario S1 is 1.35 tons $CO_2e/m^2$, which is more compared to Scenario BS. The least favorable scenario for the environment, in Phase 1, is Scenario S2 whose embodied carbon is 153.00 tons $CO_2e$, for the whole building, which is 14.60 tons $CO_2e$ more than that in Scenario BS with an increase of 10.55%. The obtained value calculated per $m^2$ for Scenario S2 is 1.39 tons $CO_2e/m^2$, which is more than Scenario BS.

The conclusion drawn from Phase 1 is that, in the short-run, Phases A1–A5 in Scenario BS have the least impact on the environment followed by Scenario S1 with greater impact and finally Scenario S2 with the greatest impact on the environment. The results from Phase 1 are short-term. The building then enters the operational phase, which was analyzed in Phase 2.

**Table 4.** Embodied carbon LCA benchmark in Phase 1 for Scenarios BS, S1 and S2.

| Analyzed Scenario | | Embodied Carbon | | | |
|---|---|---|---|---|---|
| | | Tons of $CO_2e$ per Building | Tons of $CO_2e$ per net $m^2$ | More Tons of $CO_2e$ than BS | Increase % |
| 1. | BS | 138.40 | 1.26 | 0 | 0 |
| 2. | S1 | 148.20 | 1.35 | 9.80 | 7.08% |
| 3. | S2 | 153.00 | 1.39 | 14.60 | 10.55% |

*5.3. Results of Scenarios BS, S1 and S2 upon the Completion of the Analysis of Total Carbon Footprint LCA, within Boundaries of Phases A1–B2 (after 10 Years of Building Exploitation)*

The results of total carbon footprint (embodied and operational in 10 years of building exploitation) for the analyzed scenarios were obtained after Phase 2 was completed and are shown in Table 5. These are the long-run scores of the applied measures on energy efficiency of buildings in Serbia. The grouping of the materials is in accordance with UK national standard and based on the average energy consumption in the industrial sector.

**Table 5.** Values of total carbon footprint in Phase 2 within boundaries A1–B2 after 10 years and percentage of participation of groups of materials and activities in analyzed scenarios (BS, S1 and S2).

| Groups of Materials and Activities | BS | | S1 | | S2 | |
|---|---|---|---|---|---|---|
| | Tons $CO_2e$ | % | Tons $CO_2e$ | % | Tons $CO_2e$ | % |
| Quarried material | 44.40 | 32.08 | 44.40 | 29.96 | 44.40 | 29.01 |
| Timber | 3.40 | 2.46 | 3.40 | 2.29 | 3.40 | 2.21 |
| Concrete, mortars &cement | 28.40 | 20.52 | 28.40 | 19.16 | 28.40 | 18.55 |
| Metals | 23.90 | 17.27 | 23.90 | 16.13 | 23.90 | 15.61 |
| Plastics | 5.80 | 4.19 | 5.80 | 3.91 | 5.80 | 3.79 |
| Glass | 1.40 | 1.01 | 3.70 | 2.50 | 5.60 | 3.65 |
| Miscellaneous | 1.60 | 1.16 | 9.00 | 6.07 | 11.90 | 7.77 |
| Finishing, coatings & adhesives | 7.10 | 5.13 | 7.10 | 4.79 | 7.10 | 4.64 |
| Plant and equipment emissions | 5.40 | 3.90 | 5.40 | 3.65 | 5.40 | 3.52 |
| Waste removal | 1.10 | 0.80 | 1.10 | 0.74 | 1.10 | 0.72 |
| Portable site accommodation | 2.00 | 1.45 | 2.00 | 1.35 | 2.00 | 1.31 |
| Material transport | 5.50 | 3.96 | 5.60 | 3.78 | 5.70 | 3.73 |
| Personnel travel | 8.40 | 6.07 | 8.40 | 5.67 | 8.40 | 5.49 |
| **Operational** | 47.76 | 25.65 | 15.66 | 9.56 | 7.80 | 4.65 |
| **Total Carbon Footprint** | 186.16 | 100.00 | 163.86 | 100.00 | 160.80 | 100.00 |

The percentage of the certain groups of materials and activities in Phases A1–A5 as well as the percentage of whole life carbon footprint created during the 10-year operation for Phases B1–B2 of Scenarios BS, S1 and S2 are shown in Table 5. Operational phase in the first 10 years of exploitation was included in the calculation, but maintenance, repair, replacement and reconstruction are outside the system boundaries. The total carbon footprint within the boundaries of Phase A1–B2 (during 10 years of exploitation) in Scenario BS is 186.16 tons $CO_2e$, which is the highest value among all three analyzed scenarios. The next lower total carbon footprint within the boundaries of Phases A1–B2 is Scenario S1 with the value of 163.86 tons $CO_2e$. The lowest total carbon footprint within the boundaries of Phases A1–B2 is 160.80 tons $CO_2e$, in Scenario S2.

*5.4. Discussion and Comparison of Scenarios BS, S1 and S2 upon the Completion of the Analysis of Total Carbon Footprint after 10 Years of Building Exploitation*

At the beginning of building exploitation, Scenario BS is the most favorable for the environment due to the lowest level of embodied carbon. This trend continues after the first year of building exploitation because the total value of carbon footprint for Scenarios S1 and S2 is higher than for

Scenario BS, as shown in Table 6. After 3.05 years, the values of total carbon footprint in BS and S1 reach the same values and then Scenario S1 becomes the better option for the environment. This time parameter is shown in Figure 6 (Point A). The difference of 9.80 tons $CO_2$e in embodied carbon enables Scenario BS to have lower carbon footprint for the first 3.05 years than Scenario S1. By comparing Scenario BS with Scenario S2, it can be concluded that Scenario S2 has higher carbon footprint by 14.60 tons $CO_2$e than Scenario BS. In the short-run, Scenario S2 has more impact on the environment than Scenario BS.

**Table 6.** Embodied carbon footprint LCA benchmark for Scenarios BS, S1 and S2 and achieved savings.

| Analyzed Scenario | | Embodied Carbon Footprint and Whole Life Carbon after 1 Year | | | | |
|---|---|---|---|---|---|---|
| | | Embodied Tons $CO_2$e per Building | Operational $CO_2$ | Total $CO_2$ | More Tons $CO_2$e Than BS | % Reduction $CO_2$ Total |
| 1. | BS | 138.40 | 4.774 | 143.174 | 0 | 0 |
| 2. | S1 | 148.20 | 1.565 | 149.765 | 6.591 | 4.60 |
| 3. | S2 | 153.00 | 0.78 | 153.78 | 10.606 | 7.41 |

After 3.66 years, the values of total carbon footprint in Scenarios BS and S2 reach the same values and only then Scenario S2 will become a better option for the environment than Scenario BS. The time parameter is shown in Figure 6 (Point B).

By comparing Scenario S1 with Scenario S2, it is clearly shown that Scenario S2 has higher values of embodied carbon by 4.80 tons $CO_2$e than Scenario S1. It takes 6.12 years for the two scenarios to reach the same values of total carbon footprint and, after 6.12 years of building exploitation, Scenario S2 becomes a better option for the environment than Scenario S1.

The benchmark values of the carbon footprint of the analyzed scenarios, after 10 years of exploitation, are shown in Table 7. Long-term assessment shows that, after 10 years of building operation, the least favorable option, regarding the impact on the environment, is Scenario BS which presents a building without thermal insulation layers and whose total carbon footprint, after 10 years, amounts to 186.16 tons $CO_2$e.

**Table 7.** Embodied carbon footprint LCA benchmark for Scenarios BS, S1, S2 and achieved savings after 10 years of exploitation.

| Analyzed Scenario | | Embodied and Operational Carbon Footprint after 10 Years | | | | |
|---|---|---|---|---|---|---|
| | | Embodied Tons $CO_2$e per Building | Operational $CO_2$ | Total $CO_2$ | Less Tons $CO_2$e Than (BS) | % Reduction $CO_2$ Total |
| 1. | BS | 138.40 | 47.76 | 186.16 | 0 | 0 |
| 2. | S1 | 148.20 | 15.66 | 163.86 | 22.30 | 11.98 |
| 3. | S2 | 153.00 | 7.80 | 160.80 | 25.36 | 13.62 |

In the long-run, Scenario S1 is more favorable scenario for the environment than Scenario BS, because it presents the building in energy rating C. After 10 years of building operation, the total carbon footprint in Scenario S1 is 163.86 tons $CO_2$e, which is lower by 22.30 tons $CO_2$e or 11.98% compared with Scenario BS.

After the 10-year period of building exploitation, the most favorable choice regarding the impact on the environment, measured through total carbon footprint $CO_2$e, is Scenario S2. Scenario S2 is a building in energy rating B whose total carbon footprint is 160.80 tons $CO_2$e, which is 25.36 tons $CO_2$e or 13.62% lower than Scenario BS. Embodied and whole life carbon footprint for analyzed scenarios are shown in Figures 6 and 7.

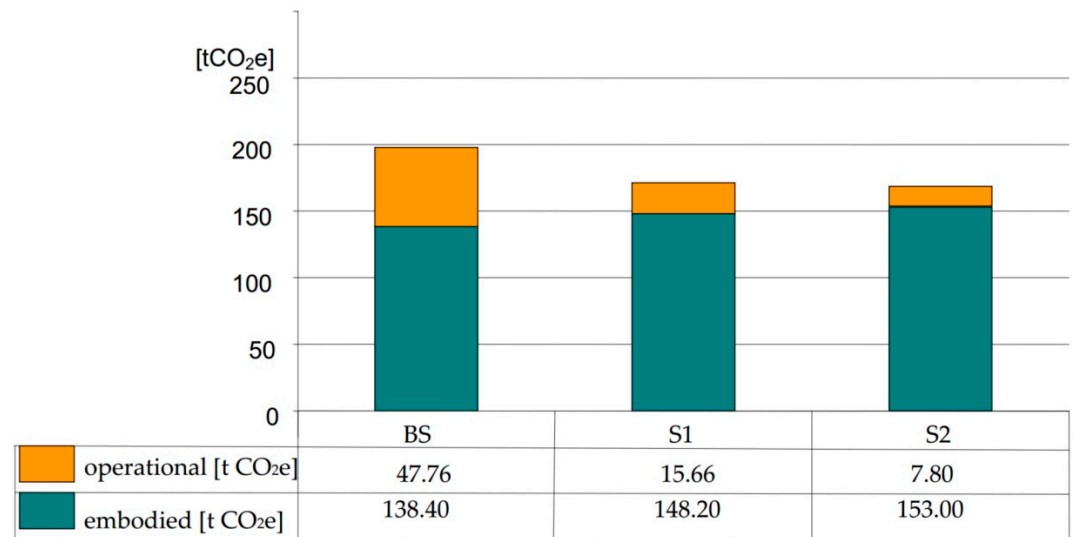

**Figure 6.** Total carbon footprint of analyzed scenarios after 10 years.

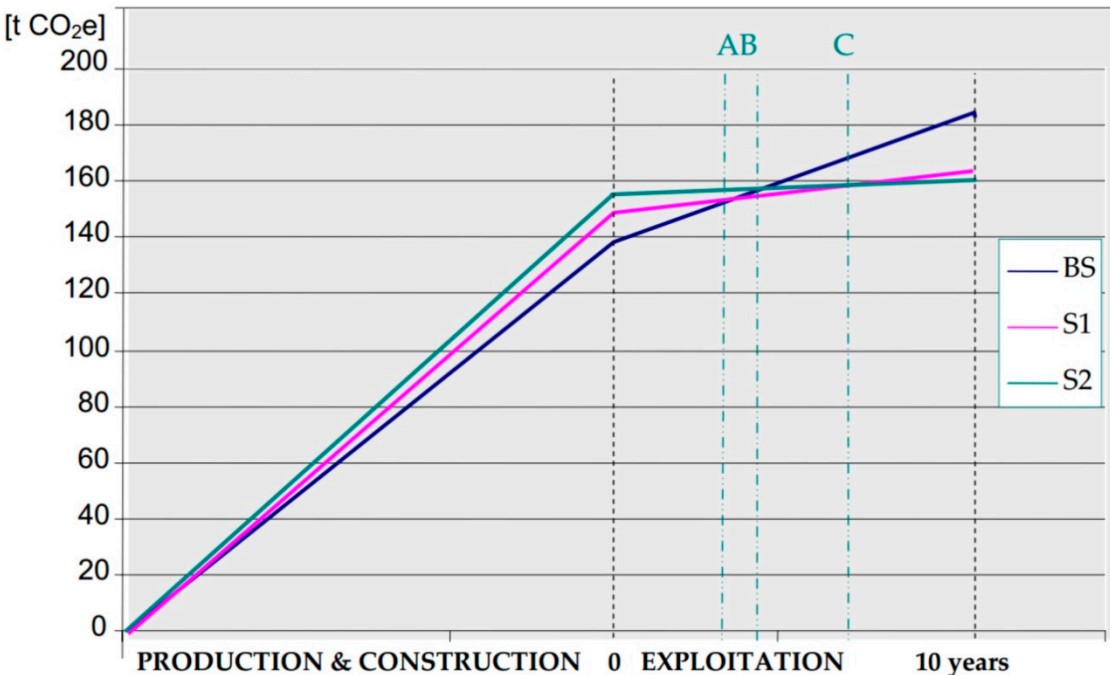

**Figure 7.** Flow of embodied and operational carbon for construction and 10-year exploitation.

Short-term assessment indicates that the most favorable scenario, regarding the impact on the environment, is Scenario BS which from construction phase and in the first couple of years of operation has the lowest impact on the environment (Figure 7). The increase in embodied $CO_2$ in Phases A1–A5, caused by higher quantity of thermal insulation materials and façade carpentry with the improved thermal features for the analyzed Scenarios S1 and S2, brings greater burden for the environment. The embodied carbon in analyzed Scenario S1 is 9.80 tons $CO_2e$ or 7.08% higher (Table 4) than Scenario BS. In addition, in the short-run, the analyzed Scenario S2 has 14.60 tons $CO_2e$ or 10.55% higher embodied carbon (Table 4) compared to Scenario BS. The increase occurred due to the installation of bigger quantities of thermal insulation materials and facade carpentry with the improved thermal features in the construction phase which causes greater burden for the environment compared to Scenario BS. However, the values of total carbon footprint for Phases A1–B2 after 10 years of building operation reveal completely different results (Figure 7). Scenario BS becomes the scenario with the

highest amount of total carbon footprint which amounts to 186.16 tons $CO_2e$, which is 20.30 tons $CO_2e$ or 11.98% higher than Scenario S1 (163.86 tons $CO_2e$). Compared with Scenario S2, whose total carbon footprint is 160.80 tons $CO_2e$, the baseline scenario is higher by 25.36 tons $CO_2e$ or 13.62%.

## 6. Conclusions

This study compares the embodied carbon in Phase 1within the boundaries of Phases A1–A5 of residential buildings in Serbia designed in a conventional way. Phase 2 compares the total carbon footprint of Phases A1–B2 including 10 years of exploitation of the building to estimate when the positive effects of the applied measures on energy efficiency in residential buildings can be expected.

Carbon footprint was analyzed in three scenarios created for the research to show the impact on the environment caused by the flow of thermal insulation materials and façade carpentry with the improved technical features. The design project and the selection of necessary materials are done for each scenario, based on which the quantities of construction materials, activities and processes participating in the construction of analyzed scenarios (BS, S1 and S2) were calculated. Referential building BS was designed without thermal insulation layers, with external carpentry with poor thermal features. The design for Scenario S1 included thermal insulation layers and improved external carpentry on every exterior construction element, so that the building fulfills the basic requirements according to the valid legislation in Serbia and that is energy rating C. The design of the building in Scenario S2 included thermal insulation layers and improved facade carpentry on every exterior construction element, to fulfill the requirements for energy rating B.

The results of Phase 1 show that Scenario BS is the most favorable in the short-run due to the lowest level of embodied carbon, which is 138.40 tons $CO_2e$. In the short-run, Scenario S1 is less favorable for the environment because of the higher quantity of the embodied carbon, which is 148.20 tons $CO_2e$. In the short-run, the scenario with the highest value of embodied carbon is Scenario S2 amounting to 153.00 tons $CO_2e$. However, if the effects of the applied measures on energy efficiency of buildings are estimated in the long-run, the score is different. The total carbon footprint for the Scenario BS after 10 years of building exploitation is the option with the highest impact on the environment, which is 186.16 tons $CO_2$. Scenario BS is the least favorable for the environment. In the long-run, Scenario S1 with the total carbon footprint of 163.86 tons $CO_2$ becomes more favorable than Scenario BS after 3.05 years. The most favorable scenario in the long-run is Scenario S2 with the total carbon footprint of 160.80 tons $CO_2$. Scenario S2 becomes a better option regarding the impact on the environment than Scenario BS after 3.66 years. Scenario S2 is a better option than Scenario S1 after 6.12 years of building exploitation.

The research shows that it is necessary to include the calculation of embodied carbon in the design stage and not only operational carbon when deciding the energy rating of a building. The study shows the necessity for the low-carbon construction materials, especially thermal insulation, which are gaining more importance regarding the reduction of energy needs in a building. The results indicate that the disclosure of embodied $CO_2$ clarifies the possibilities for the reduction of impact on the environment from the construction sector as well as the expected effects from the regulations in that field in Serbia.

Further studies in this field are to be directed towards the comparison of the existing thermal insulation materials with regards to their impact on the calculation of embodied and operational carbon footprint to estimate their life span, i.e., all exploitation time and replacement frequency. Future studies could also include the research on the length of transportation route, i.e., how the mode of transportation affects the embodied carbon. Another type of multidisciplinary research should include research on finding low-carbon thermal insulation materials in national construction industry which will help bridge the gap between the growing need for thermal insulation materials and the need for the reduction of $CO_2$ emissions.

**Author Contributions:** Each author participated and contributed sufficiently to take public responsibility for appropriate portions of the content.

**Funding:** This research received no external funding.

**Conflicts of Interest:** The authors declare no conflict of interest.

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
