# Peer review of "Comparison of the Applied Measures on the Simulated Scenarios for the Sustainable Building Construction through Carbon Footprint Emissions—Case Study of Building Construction in Serbia"

_sustainability, doi:10.3390/su10124688_

Round 1

Reviewer 1 Report

The paper presents an applied measures assessment for the sustainable building construction according carbon footprint in Serbia. Authors indicate the impact of increased flow of thermal materials on environment regarding an implementation of new regulations on buildings energy efficiency. Carbon footprint was analyzed in three scenarios for which the quantities of construction materials and processes are calculated. Analysis of life cycle was estimated, which is the basis for analyzing the carbon life cycle. Authors used carbon calculator for measurement of carbon footprint. The research was done in two steps. Embodied carbon was measured to evaluate short‐term effects of new regulations. Obtained results show that in the short‐run, the scenario (BS) has the smallest embodied carbon. In case of long-run, after 3,66 years scenario (S2) becomes a better option according an impact on the environment. The study indicates the necessity to include embodied carbon with whole life carbon to evaluate the impact of buildings on the environment.

Authors say about new regulations? Is it European regulations or Serbian – domestic regulations? It is important to know for a readers.

32 – introduction title should be moved to next page.

I think, Introduction is correct. Mentioned references and their number is also correct. I would expect more technical details in introduction respectively to type of journal – Sustainability Journal.

62 – please decide what is capital letter and what is not in chapter title. Just first word or all of them. See also other chapter titles.

108-109 – the energy consumption in case of residential and office buildings kWh/m2 is for one year or total life of the building? Please clarify it.

Chapter 2 describes in details investigated problem. The topic is very important, especially in case of EU regulations and problem with CO2. Usually, CO2 is associated to energy and power sector. But the true is that there are many other resources of this gas.

Chapter Methodology is too long and there are too many information. In my mind some information are superfluous and make a reader confused.

405 – table 2 should be moved to next page to be together with whole table in one page. Please correct it.

429, 448, 469 – see comments to line 405.

431 – Fig.4 – I think, axis Y should be described with unit. Also values on the axis are difficult to read. Please correct it.

485 – see comments to line 431.

485 – description of Fig.5 should be together with picture. Please correct it.

486 – see comments to line 431.

Author Response

Dear Editor,

Thank you very much for the useful suggestions. We accepted all of the suggestions and we are sure that his will improve the quality and contribute to a better understanding of the paper.

At the request of the reviewers, the following corrections have been made

Reviewer 1:

Comment 1: Authors say about new regulations? Is it European regulations or Serbian – domestic regulations? It is important to know for a readers.

Response to comment 1:

Thank you for this suggestion. The regulations listed in this article are Serbian, but are also in correlation to European. We corrected this in the new version of the paper.

Comment 2: In lines 32 – introduction title should be moved to next page.

Response to comment 2:

We corrected this in the new version of the paper.

Comment 3: I think, Introduction is correct. Mentioned references and their number is also correct. I would expect more technical details in introduction respectively to type of journal – Sustainability Journal.

Response to comment 3:

Thank you for this suggestion. We added references of journal Sustainability Journal. We corrected this in the new version of the paper.

Comment 4: In lines 62 – please decide what is capital letter and what is not in chapter title. Just first word or all of them. See also other chapter titles.

Response to comment 4:

We corrected this in the new version of the paper.

Comment 5: In lines 108-109 – the energy consumption in case of residential and office buildings kWh/m2 is for one year or total life of the building? Please clarify it.

Response to comment 5: The energy consumption of residential and office buildings in this case is for one year. Corrected in the new version of the paper.

Comment 6: Chapter 2 describes in details investigated problem. The topic is very important, especially in case of EU regulations and problem with CO2. Usually, CO2 is associated to energy and power sector. But the true is that there are many other resources of this gas.

Response to comment 6: We agree with your views, the embodied CO2 potions partly from the consumed energy in the production, transport, processing and construction.

Comment 7: Chapter Methodology is too long and there are too many information. In my mind some information are superfluous and make a reader confused.

Response to comment 7: Thanks to this suggestion, some parts of this chapter have been removed. Corrected in the new version of the paper.

Comment 8: In lines 405 – table 2 should be moved to next page to be together with whole table in one page. Please correct it.

Response to comment 8: Corrected in the new version of the paper.

Comment 9: In lines 429, 448, 469 – see comments to line 405.

 Please correct it.

Response to comment 9: Corrected in the new version of the paper

Comment 10: In lines 431 – Fig.4 – I think, axis Y should be described with unit. Also values on the axis are difficult to read. Please correct it.

Response to comment 10:  Figure 4 is corrected in the new version of the paper.

Comment 11: In lines 485 – see comments to line 431.

Response to comment 11: Corrected in the new version of the paper.

Comment 12: In lines 485 – description of Fig.5 should be together with picture. Please correct it.

Response to comment 12: Corrected in the new version of the paper.

Comment 13: In lines 486 – see comments to line 431

Response to comment 13: Corrected in the new version of the paper.

Reviewer 2 Report

English editing is necessary, as some of the sentences are difficult to understand. Quality of Fig. 4 must be improved.

Author Response

Dear Editor,

Thank you very much for the useful suggestions. We accepted all of the suggestions and we are sure that his will improve the quality and contribute to a better understanding of the paper.

At the request of the reviewers, the following corrections have been made

Reviewer 2:

Comment 1: Moderate English changes required

Response to comment 1: Thank you for this suggestion, we hope the changes will enhance the quality of this paper.  Corrected in the new version of the paper.

Comment 2: Quality of Fig. 4 must be improved.

Response to comment 2: Quality of Fig. 4 is improved. Corrected in the new version of the paper.

Comments and Suggestions for Authors: English editing is necessary, as some of the sentences are difficult to understand.

Response to comments and Suggestions  Sentences, which due to their length were not clear enough, were reformulated. Corrected in the new version of the paper.

Reviewer 3 Report

Title:

It is necessary to indicate in the title the term: comparative of simulated cases

Abstract:

I do not know if it will be specified later, but 10 years as a horizon to determine the impact of the operational energy is very limited (the life of buildings is usually estimated for 50 years, and the reality is that in many cases it is 100 years).

The issue of energy efficiency is not an innovative issue, in Europe these regulations and energy improvement processes have been regulated for more than 10 years, with regulations and with real application. In fact, at present, assuming guaranteed or optimized operational energy, the new paradigms in LCA focus on the embodied energy of the materials (stages of production, construction); and even less exploratory, are Maintenance, Repairing, Replacement, Renovation, re-use, recycling, etc. It is considered necessary to reflect on the fundamental structure of the work.

Keywords:

It is considered that LCA should be included as a keyword

Introduction

I would suggest to the authors to review this section, in order to achieve a "complete and general vision" of what is intended in this work. The current content presented is generic.

Summary of the reference documents used for the investigation:

Usually, for energy efficiency issues, the solutions that are usually recommended or improvements (for simplicity, feasibility and economy) are the facilities

It is suggested to the authors to review this work for treating LCA in phases of production-transport and construction of constructive elements:

https://www.mdpi.com/2071-1050/10/8/2748

The tessitura of the presented summaries are focused to expose practical cases or experiences of applications; however, there is not enough information about the incorporated energy (evaluation criteria, experimentation procedures, existing regulations, etc.). It is suggested that the authors include information directly related to the central content of this research.

Please highlight the innovation that this work will explore, the difference it has with respect to what has been investigated so far, the significant scientific contribution that is made, etc.

Methodology

Figure 1. For any LCA researcher or professional in the construction industry, this figure is known, does not provide new information, and therefore the authors are asked to eliminate it and simply indicate the bibliographic reference.

Please, do not explain the 14040: 2006 regulation, it is known by all researchers; It is requested that it be explained directly how it is put into practice in this investigation. It is necessary to rewrite this section avoiding duplicating existing information, explaining what is defined in the regulations, or exposing information not necessary or essential for this investigation.

Please, do not explain the regulation EN 15978: 2011, it is known by all researchers; It is requested that it be explained directly how it is put into practice in this investigation.

Figure 2 and 3. They are considered excessive; it is better to simply indicate in the text. It is requested to eliminate them for not providing significant information for this investigation.

Explain the reasons why maintenance, repair, renovation, end of life, re-use and recycling are not included. It is considered that it would be of interesting scientific contribution if the authors include these stages. Your inclusion in the study is requested.

Justify that only 10 years of study are included (it is considered small and unrealistic)

It is considered that the use of databases or programs of limited use (not available) is not adequate. In specific the reference [55] is not available -31/10/2018-. Authors are only allowed to use internationally accepted databases (for the European scope: Ecoinvent). The calculator used, reference [56], is not available -31/10/2018-. Authors are asked to use available programs. The URSA program, reference [57] is not available -31/10/2018-. Authors are requested to use available computer programs. It is mandatory criteria, that all processes, calculations, databases and other information used or process tool of it are available, other researchers MUST be able to replicate the investigation. This criterion is mandatory to be able to accept publishing this work.

It is necessary that the authors argue that the study building is representative, that it has statistical solvency to be able to infer results extrapolated to the knowledge of this topic, etc.; otherwise, it is only a case of study and therefore its conclusions or contributions have no interest for other researchers.

Indicate the database used to carry out the inventory (must be available for consultation with other researchers).

Case study

The content of this paragraph does not respond to the indicated title. It is requested to review. It is expected that in this title the data and information of the case study will be exposed.

Description of the experiment

It is necessary to include architectural graphic documentation of the case study.

S1 and S2: It is necessary to include graphic details of the constructive solutions proposed in these scenarios.

Table 1. If the materials used to carry out the inventory for the cases BS, S1 and S2 are identical (for example, for Tamping gravel = 75.00) The authors consider that it can be a constant in the three evaluation cases and therefore can skip The objective of this research is the comparison of the three cases, not the global affectation. So if we studied only the different inventories we could not only distinguish the problem on a real scale, but also identify the origin of the variation (therefore, the recommendation or improvements established in the conclusions would be simpler)

Replace quantities

Authors are asked not to repeat previous information that already included in the work. Review and eliminate duplicate information.

Question: Is gypsum prescribed for exterior walls on both sides?

It is necessary to indicate the properties of each constructive element (transmittance, fire resistance, geometry, density, etc.), compliance with the standards (what regulations they comply with)

The authors are asked to explain: the constructive solutions used in S1 and S2 are improvements of the SB study case, if the answer is yes, then the improvement solutions are feasible to be carried out in the SB building (technically, regulation, legislation, etc.). If not, and the improvements of the S1 and S2 buildings imply that they are incompatible with the existing SB building, it means that they are different buildings and therefore IT IS NOT POSSIBLE TO REALIZE YOUR REAL COMPARATIVE. Therefore, it is necessary to demonstrate that the improvements S1 and S2 have a precedent to SB and are feasible to perform (constructively).

It is necessary to improve all the data in this section, rewrite and improve the information.

Results and Discussion

Table 2, 3 and 4. It is recommended to carry out a more detailed analysis and explain it in the text (origin of the problem, constructive systems involved, implications, extrapolations of data, etc.)

If the information in Table 3 and 4 is the same as that in Figure 4 and 5 (respectively), it is requested to simplify (avoid duplicating the information). It is recommended to leave the graph alone for allowing a simpler perception of relationships.

Improve the vertical axis scale of Figure 4 and 5 (limit to 250 tCO2e)

The analysis that is exposed of the values does not go beyond a comparative in subjective terms. A deeper analysis is requested, as well as explaining the origin of the variations and their implications.

Figure 6. It should be better explained and it is recommended to improve the format of the same.

Conclusion:

The authors are asked to rewrite this section taking into account all the previous comments.

Bibliography:

It is recommended to perform an exhaustive search of the last 10 years in databases: for example: SCOPUS, Compendex or WoS

Author Response

Dear Editor,

Thank you very much for the useful suggestions. We accepted majority of the suggestions and we are sure that his will improve the quality and contribute to a better understanding of the paper.

At the request of the reviewers, the following corrections have been made

Reviewer 3:

Comment 1:Title:It is necessary to indicate in the title the term: comparative of simulated cases

Response to comment 1:
Thank you for this suggestion, the title has been changed. Corrected in the new version of the paper.

Comment 2: Abstract:

I do not know if it will be specified later, but 10 years as a horizon to determine the impact of the operational energy is very limited..

Response to comment 2: Thank you for this suggestion,
in the further course of work, an additional explanation is given why only the first 10 years of operational energy is considered.Corrected in the new version of the paper.

Comment 3: Abstract:

The issue of energy efficiency is not an innovative issue, in Europe these regulations and energy improvement processes have been regulated for more than 10 years, with regulations and with real application. In fact,

Response to comment 3:
Thank you for suggesting, but in Serbia there is legislation on energy efficiency of a newer date. It has been adopted and is being applied since September 2012, and it is important for this region to consider how much the embodied carbon increased due to the implementation of 2012 measures. This research has been done to show that the methodology for calculating energy rating building in Serbia needs to undergo changes. This is better explained in the paper. Corrected in the new version of the paper.

Comment 4: Keywords:

It is considered that LCA should be included as a keyword

Response to comment 4: Thank you for this suggestion. Corrected in the new version of the paper.

Comment 5: Introduction

I would suggest to the authors to review this section, in order to achieve a "complete and general vision" of what is intended in this work. The current content presented is generic.

Response to comment 5: Thank you for this suggestion. Corrected in the new version of the paper.

Comment 6: Summary of the reference documents used for the investigation:

Usually, for energy efficiency issues, the solutions that are usually recommended or improvements (for simplicity, feasibility and economy) are the facilities

It is suggested to the authors to review this work for treating LCA in phases of production-transport and construction of constructive elements:

https://www.mdpi.com/2071-1050/10/8/2748

Response to comment 6: Thank you for this suggestion. Corrected in the new version of the paper.

Comment 7: The tessitura of the presented summaries are focused to expose practical cases or experiences of applications; however, there is not enough information about the incorporated energy (evaluation criteria, experimentation procedures, existing regulations, etc.). It is suggested that the authors include information directly related to the central content of this research.

Response to comment 7: Thank you for this suggestion. Corrected in the new version of the paper.

Comment 8: Please highlight the innovation that this work will explore, the difference it has with respect to what has been investigated so far, the significant scientific contribution that is made, etc.

Response to comment 8: Thank you for this suggestion. Corrected in the new version of the paper.

Comment 9: Methodology, Figure 1. For any LCA researcher or professional in the construction industry, this figure is known, does not provide new information, and therefore the authors are asked to eliminate it and simply indicate the bibliographic reference.

Response to comment 9: Thank you for this suggestion. Corrected in the new version of the paper.

Comment 10: Please, do not explain the 14040: 2006 regulation, it is known by all researchers; It is requested that it be explained directly how it is put into practice in this investigation. It is necessary to rewrite this section avoiding duplicating existing information, explaining what is defined in the regulations, or exposing information not necessary or essential for this investigation.

Response to comment 10: Thank you for this suggestion. Corrected in the new version of the paper.

Comment 11: Please, do not explain the regulation EN 15978: 2011, it is known by all researchers; It is requested that it be explained directly how it is put into practice in this investigation.

Response to comment 11: Thank you for this suggestion. Corrected in the new version of the paper.

Comment 12: Figure 2 and 3. They are considered excessive; it is better to simply indicate in the text. It is requested to eliminate them for not providing significant information for this investigation.

Response to comment 12: Thank you for this suggestion. Corrected in the new version of the paper.

Comment 13: Explain the reasons why maintenance, repair, renovation, end of life, re-use and recycling are not included. It is considered that it would be of interesting scientific contribution if the authors include these stages. Your inclusion in the study is requested. Justify that only 10 years of study are included (it is considered small and unrealistic)

Response to comment 13:
These phases are not included because the research wants to identify the lack of current regulations, in the sense that the embodied carbon is not included in the calculation of the object's savings in real terms, the environment and neglect of the emissions that arise from A1-A5, when calculating emissions from buildings. We believe that this period is sufficient from the previously presented point of view, new materials and new regulations will arrive, and in this sense new methods for calculation. The work is directed to look at how the regulations for assessing the impact of the facility on the environment should be corrected. Corrected in the new version of the paper.

Comment 14: It is considered that the use of databases or programs of limited use (not available) is not adequate. In specific the reference [55] is not available -31/10/2018-. Authors are only allowed to use internationally accepted databases (for the European scope: Ecoinvent). The calculator used, reference [56], is not available -31/10/2018-. Authors are asked to use available programs. The URSA program, reference [57] is not available -31/10/2018-. Authors are requested to use available computer programs. It is mandatory criteria, that all processes, calculations, databases and other information used or process tool of it are available, other researchers MUST be able to replicate the investigation. This criterion is mandatory to be able to accept publishing this work.

Response to comment 14:
Thank you for this suggestion. Programs that have been used are free accessible and are available to users in Serbia. Internet addresses have been corrected. Corrected in the new version of the paper.

Comment 15: It is necessary that the authors argue that the study building is representative, that it has statistical solvency to be able to infer results extrapolated to the knowledge of this topic, etc.; otherwise, it is only a case of study and therefore its conclusions or contributions have no interest for other researchers.

Response to comment 15: Thank you for this suggestion. Corrected in the new version of the paper.

Comment 16: Indicate the database used to carry out the inventory (must be available for consultation with other researchers).

Response to comment 16: Thank you for this suggestion. Corrected in the new version of the paper.

Comment 17: Case study

The content of this paragraph does not respond to the indicated title. It is requested to review. It is expected that in this title the data and information of the case study will be exposed.

Response to comment 17: Thank you for this suggestion. Corrected in the new version of the paper.

Comment 18: Description of the experiment

It is necessary to include architectural graphic documentation of the case study.

.

Response to comment 18: Thank you for this suggestion. Corrected in the new version of the paper.

Comment 19: S1 and S2: It is necessary to include graphic details of the constructive solutions proposed in these scenarios.

Response to comment 19: Thank you for this suggestion. Corrected in the new version of the paper.

Comment 20: Table 1. If the materials used to carry out the inventory for the cases BS, S1 and S2 are identical (for example, for Tamping gravel = 75.00) The authors consider that it can be a constant in the three evaluation cases and therefore can skip The objective of this research is the comparison of the three cases, not the global affectation. So if we studied only the different inventories we could not only distinguish the problem on a real scale, but also identify the origin of the variation (therefore, the recommendation or improvements established in the conclusions would be simpler).

Response to comment 20: Thank you for this suggestion.
However, we believe that it is necessary to give all the quantities that are within the limits of the system in order to get a clear picture of the embodied carbon of each of the analyzed scenarios.

Comment 21: Replace quantities

Authors are asked not to repeat previous information that already included in the work. Review and eliminate duplicate information.

Response to comment 21: Thank you for this suggestion. Corrected in the new version of the paper.

Comment 22: Question: Is gypsum prescribed for exterior walls on both sides?

Response to comment 22: Thank you for this suggestion. Gypsum was not used, it can be seen in the accompanying wall drawings. Corrected in the new version of the paper.

Comment 23: It is necessary to indicate the properties of each constructive element (transmittance, fire resistance, geometry, density, etc.), compliance with the standards (what regulations they comply with)

Response to comment 23: Thank you for this suggestion. Corrected in the new version of the paper.

Comment 24: The authors are asked to explain: the constructive solutions used in S1 and S2 are improvements of the SB study case, if the answer is yes, then the improvement solutions are feasible to be carried out in the SB building (technically, regulation, legislation, etc.). If not, and the improvements of the S1 and S2 buildings imply that they are incompatible with the existing SB building, it means that they are different buildings and therefore IT IS NOT POSSIBLE TO REALIZE YOUR REAL COMPARATIVE. Therefore, it is necessary to demonstrate that the improvements S1 and S2 have a precedent to SB and are feasible to perform (constructively).

Response to comment 24: Thank you for this suggestion.
The constructive system for all three scenarios is the same, only the difference in the energy class scenarios. Corrected in the new version of the paper.

Comment 25: It is necessary to improve all the data in this section, rewrite and improve the information.

Response to comment 25: Thank you for this suggestion. Corrected in the new version of the paper.

Comment 26: Results and Discussion

Table 2, 3 and 4. It is recommended to carry out a more detailed analysis and explain it in the text (origin of the problem, constructive systems involved, implications, extrapolations of data, etc.)

Response to comment 26: Thank you for this suggestion. Corrected in the new version of the paper.

Comment 27: If the information in Table 3 and 4 is the same as that in Figure 4 and 5 (respectively), it is requested to simplify (avoid duplicating the information). It is recommended to leave the graph alone for allowing a simpler perception of relationships.

Response to comment 27: Thank you for this suggestion. Corrected in the new version of the paper.

Comment 28: Improve the vertical axis scale of Figure 4 and 5 (limit to 250 tCO2e)

Response to comment 28: Quality of Fig. 4 and 5 is improved. Corrected in the new version of the paper.

Comment 29: The analysis that is exposed of the values does not go beyond a comparative in subjective terms. A deeper analysis is requested, as well as explaining the origin of the variations and their implications.

Response to comment 29: Thank you for this suggestion. Corrected in the new version of the paper.

Comment 30: Figure 6. It should be better explained and it is recommended to improve the format of the same.

Response to comment 30: Thank you for this suggestion. Corrected in the new version of the paper.

Comment 31: Conclusion:

The authors are asked to rewrite this section taking into account all the previous comments.

Response to comment 31: Thank you for this suggestion. Corrected in the new version of the paper.

Comment 32: Bibliography:

It is recommended to perform an exhaustive search of the last 10 years in databases: for example: SCOPUS, Compendex or WoS

Response to comment 32: Thank you for this suggestion. Corrected in the new version of the paper.

We hope that the corrections that have been made will enhance the quality of this paper. Corrected in the new version of the paper.

Round 2

Reviewer 1 Report

The paper presents an applied measures assessment for the sustainable building construction according carbon footprint in Serbia. Authors indicate the impact of increased flow of thermal materials on environment regarding an implementation of new regulations on buildings energy efficiency. Carbon footprint was analyzed in three scenarios for which the quantities of construction materials and processes are calculated. Analysis of life cycle was estimated, which is the basis for analyzing the carbon life cycle. Authors used carbon calculator for measurement of carbon footprint. The research was done in two steps. Embodied carbon was measured to evaluate short‐term effects of new regulations. Obtained results show that in the short‐run, the scenario (BS) has the smallest embodied carbon. In case of long-run, after 3,66 years scenario (S2) becomes a better option according an impact on the environment. The study indicates the necessity to include embodied carbon with whole life carbon to evaluate the impact of buildings on the environment.

All my comments are included in new version of the paper. I think, the paper is ready to be published in present version.

Author Response

Dear Editor,

Thank you very much for the useful suggestions. We accepted all of the suggestions and we are sure that his will improve the quality and contribute to a better understanding of the paper.

At the request of the reviewers, the following corrections have been made

Reviewer 1:

Comment 1: Moderate English changes required

Response to comment 1: Thank you for this suggestion, we hope the changes will enhance the quality of this paper.  Corrected in the new version of the paper.

Due to additional requests received from Reviewer 3, additional changes were made. We hope that the corrections that have been made will enhance the quality of this paper. Corrected in the new version of the paper.

Reviewer 3 Report

The requested improvements have been made. The work can be published.

Congratulations